# Air Permeability, Shock Absorption Ability, and Flexural Strength of 3D-Printed Perforated ABS Polymer Sheets with 3D-Knitted Fabric Cushioning for Sports Face Guard Applications

**DOI:** 10.3390/polym13111879

**Published:** 2021-06-05

**Authors:** Thet Khaing Aung, Hiroshi Churei, Gen Tanabe, Rio Kinjo, Kaito Togawa, Chenyuan Li, Yumi Tsuchida, Phyu Sin Tun, Shwe Hlaing, Hidekazu Takahashi, Toshiaki Ueno

**Affiliations:** 1Department of Sports Medicine/Dentistry, Graduate School of Medical and Dental Sciences, Tokyo Medical and Dental University, 1-5-45 Yushima, Bunkyo-ku, Tokyo 113-8549, Japan; tka.spmd@tmd.ac.jp (T.K.A.); chu.spmd@tmd.ac.jp (H.C.); gen.spmd@tmd.ac.jp (G.T.); r.kinjo.spmd@tmd.ac.jp (R.K.); kaito321.629@gmail.com (K.T.); licyspmd@tmd.ac.jp (C.L.); 2Department of Maxillofacial Prosthetics, Graduate School of Medical and Dental Sciences, Tokyo Medical and Dental University, 1-5-45 Yushima, Bunkyo-ku, Tokyo 113-8549, Japan; 3Department of Oral Biomaterials Engineering, Graduate School of Medical and Dental Sciences, Tokyo Medical and Dental University, 1-5-45 Yushima, Bunkyo-ku, Tokyo 113-8549, Japan; yumi.bmoe@tmd.ac.jp (Y.T.); takahashi.bmoe@tmd.ac.jp (H.T.); 4Moe Myitta Dental Clinic, No.117, Corner of 26th x 76th Street, Chaayetharsan Township, Mandalay 05024, Myanmar; mgphyusintun@gmail.com; 5Department of Prosthodontics, The University of Dental Medicine, Mandalay, 62nd Street, Chan Mya Thazi Township, Mandalay 05041, Myanmar; drshlaingdental@gmail.com

**Keywords:** face guard, 3D-printed polymer, 3D-knitted fabric, flexural strength, shock absorption ability, air permeability, wearing comfort

## Abstract

Sports face guards (FGs) are devices that protect athletes from maxillofacial injury or ensure rapid return to play following orofacial damage. Conventional FGs are uncomfortable to wear owing to stuffiness caused by poor ventilation and often slip off due to increase in weight due to absorption of moisture from perspiration, lowering players’ performance. Herein, combinations of 3D-printed perforated acrylonitrile butadiene styrene (ABS) polymer sheets and 3D-knitted fabrics with honeycomb structures as cushioning materials were investigated to balance better wearing feel and mechanical properties. The flexural strength, weight, and shock absorption ability of, and air flow rate through, the ABS sheets with five different perforation patterns were evaluated and compared with those of conventional FG materials comprising a combination of polycaprolactone sheets for the medical splint and polychloroprene rubber for the cushioning material. The ABS sheets having 10% open area and 2.52 mm round holes, combined with knitted fabric cushioning, exhibited the requisite shock absorbing, higher air permeability, and lower weight properties than the conventional materials. Our results suggest that FGs fabricated using combinations of 3D-printed perforated ABS polymer sheets and 3D-knitted fabrics with honeycomb structures may impart enhanced wearing comfort for athletes.

## 1. Introduction

Face guards (FGs) are widely recognized as being effective in protecting the maxillofacial region during sports [1,2,3,4]. FGs can ensure the safety of athletes by minimizing maxillofacial injuries and protecting existing ones [2,3,4,5,6,7,8,9,10,11,12]. All FGs must meet the following requirements: (i) protect the player from reinjury (protective ability), (ii) not injure other players (safety), and (iii) allow the player to have a good field of vision (not hinder player performance) [5]. According to the “Laws of the Game” laid down by the International Football Association Board, the first two requirements can be described as follows: “a player may use equipment other than the basic equipment, provided that the sole purpose is to provide physical protection and that not be dangerous to the player wearing it or any other player” [13]. Thus, any adverse effect on the field of vision of the player must be minimized so that the performance of the player is not affected (requirement (iii)). This must be confirmed by evaluating the clinical effectiveness of the FG design through visual field tests [14,15].

A survey of players using FGs found that they were satisfied with the protective ability of the FGs but not with their degree of comfort, claiming that the FGs slipped while playing and narrowed their field of vision. Therefore, thinner and lighter FGs with better air permeability are required. Professional players also insist on improvements to the field of vision, bulkiness, and wearing comfort of FGs [12,15]. By combining hard thermoplastic materials and soft cushioning materials, it should be possible to realize excellent shock absorption characteristics. To further improve the capacity of FGs for shock absorption, lining the inner surface of the hard thermoplastic material with the cushioning material is a more effective strategy than placing the cushioning material on the outer surface of the thermoplastic material. Thus, while fabricating FGs, both the core material and the cushioning material are important for ensuring high shock absorption ability and comfort [11,12,13,15,16]. However, there have been no efforts to improve the wearing comfort of FGs by focusing on their air permeability.

While rubber-based materials were used as the cushioning materials in a previous study, their air permeability was not considered [17]. Three-dimensional (3D) knitted fabrics have a cushioning effect because of their stable structure, lightweight nature, good air permeability, good compression properties, and resilient behavior [18]. They are used in anti-bedsore mattresses as well as the mattresses used in wheelchairs [19]. 3D-knitted fabrics are also used for the shoulder pads in sportswear owing to their high thermal comfort, impact protection, and good air permeability [20]. Therefore, in this study, we examined the suitability of 3D-knitted fabrics for use as cushioning materials in FGs.

To improve the comfort of FGs, that is, to ensure they are lightweight in nature and possess good air ventilation characteristics, Churei et al. have examined the shock absorption abilities of several commercially available materials (medical splint sheets) with perforations (19%). They found that the perforated materials showed almost the same shock absorption abilities as those without holes [21]. Perforated materials have been explored for use as the outer-layer materials in order to reduce FG weight. The weight of the FG is an important factor affecting comfort. Although a perforated structure is considered to be effective for ensuring good air permeability, only a few studies have investigated the effect of structure on air permeability. In addition, the effects of such a structure on the strength have not been considered. Moreover, while the shock absorption abilities and other properties of these materials were evaluated without subjecting the materials to thermoforming, there are valid concerns that perforated thermoplastic materials may shrink during heat molding, resulting in reduced air permeability. To ensure both good air permeability and high strength, the precision manufacturing of the perforated structure may be necessary. For this, the use of digital technology would be highly beneficial.

The use of digital technology in the medical field is increasing rapidly because of its low technique sensitivity, high accuracy, and reduced usage time. Fixed prostheses and indirect dental restorations can now be fabricated from different materials using computer-aided design and computer-aided manufacturing technologies [22]. In addition, the shock absorption abilities of photopolymer composite materials suitable for use in 3D-printed sports mouthguards have been examined recently, and mouthguards have been fabricated successfully from these materials [23]. Digital technology reduces the number of clinical and laboratory steps, leading to fast and effective delivery of the final custom-made medical device [24,25,26,27,28,29,30,31,32].

The American Society for Testing and Materials group (ASTM F42—additive manufacturing) has formulated a set of standards that classify the range of additive manufacturing processes into seven categories: vat polymerization, material jetting, binder jetting, material extrusion, powder bed fusion, sheet lamination, and directed energy deposition [33]. There is a close relationship between the accuracy and cost of 3D printed objects, and Polyjet Matrix (PJM) technology is known to be more accurate but more costly than Fused Deposition Modeling (FDM) technology. FG, as well as being medical splints, are orthotic devices worn on the external surface of the body; hence, they need not be as highly biocompatible as artificial organs and oral devices. Based on the balance of accuracy, cost, and required biocompatibility, FDM technology was considered to be preferable for printing FG [34].

Acrylonitrile butadiene styrene (ABS) is an impact-resistant amorphous polymer that is used for 3D printing. It is composed of three monomers: acrylonitrile, butadiene, and styrene and has several desirable physical properties, such as high rigidity, good impact resistance even at low temperatures, good insulating characteristics, good weldability, and high abrasion and strain resistances [35,36]. Its high impact resistance makes it suitable for use in the automobile, electronics, building and construction, and transportation industries as well as home appliances [37,38]. In the medical industry, ABS is used to produce medical devices such as respiratory devices, infusion systems, autoinjector devices, and various other structures, such as housings for medical devices. ABS polymers are also used to produce miniature implants, such as middle ear prostheses [39,40,41]. We assumed that, even with perforations, ABS would be suitable for use as the outer-layer material for FGs because of its high strength and impact resistance.

In this study, we evaluated the shock absorption abilities and air permeabilities of various types of perforated 3D-printed ABS materials when used in combination with a 3D-knitted fabric (cushioning material) and compared them with those of commercially available materials, namely, those used for medical splints.

## 2. Materials and Methods

The conventional thermoplastic material used was Aquaplast (AP, polycaprolactone, Homecraft Rolyan, Huthwaite, North Nottingham, UK). We used 3.2-mm-thick sheets of AP without holes (labeled as AP0). The base 3D-printing material used was a 3-mm-thick ABS polymer (NCI Sales Inc., Tokyo, Japan). The cushioning materials used were neoprene (NEO, Polychloroprene with nylon lining, Homecraft Rolyan) and AKE64140 (AKE, 3D-knitted fabric, Asahi-Kasei Co., Osaka, Japan). A total of 12 combinations of the materials were used, and the specimens in all groups were subjected to shock absorption and air permeability tests.

### 2.1. Sample Preparation

The ABS sheets used for the shock absorption and air permeability tests were designed to have dimensions of 100 × 100 × 3 mm^3^. For the design process, we used the software Geomagic Freeform (version 12, 3D Systems, Co., Littleton, CO, USA). The design data were stored as Standard Tessellation Language (STL) files (Figure 1). The mesh model for conversion to STL format was automatically performed by the software. The results of conversion of each specimen are shown in Table 1 and the STL file view, and framework are shown in Figure 1. The different types of ABS sheets were labeled as follows: ABS0 (without perforations), ABS10 (10% open area and 3.64 mm rounded holes), ABS10S (10% open area and 2.52 mm rounded holes), ABS20 (20% open area and 3.57 mm rounded holes), and ABS20S (20% open area and 2.55-mm rounded holes) (Figure 2 and Table 2). The holes in the sheets were arranged in a 45° staggered formation.

The ABS sheet specimens were printed using a FDM technology of 3D printer (MyDO200, NCI Sales Inc., Tokyo, Japan) with a nozzle hole diameter of 0.4 mm. The temperature of the nozzle during printing was 230 °C for the first layer and 220 °C for the second and subsequent layers. The temperature of the printing table was 105 °C. The print pitch was 0.2 mm with a 100% rate of infill. The basic printing speed was 4000 mm/min.

### 2.2. Weight Measurements

The weights of all the specimens were measured using a weight measuring device (UH-3201-BL, A&D Co., Saitama, Japan) in room temperatures at 20–25 °C. Each measurement was repeated five times for each specimen and the mean value was reported for statistical analysis.

### 2.3. Three-Point Bending Test

The specimens for the three-point bending test (100 mm in length and 15 mm in width) were prepared using an ultrasonic cutter and waterproof abrasive paper. After the specimens had been measured with a micrometer (293-421-20, Mitsutoyo, Kanagawa, Japan; minimum reading: 0.001 mm), they were subjected to the three-point bending test using a universal test machine (Model 1123, Instron, Canton, MA, USA) with a support span width of 50 mm and crosshead speed of 1.0 mm/min. The flexural strength was calculated with a statistical software (Series IX, Instron) using the following Equation:Flexural strength = 3*Fl*/2*bh*^2^(1)
where *F* is the maximum load (N), *l* is the width of the support span (mm), *b* is the width (mm) of the specimen, and *h* is the height (mm) of the specimen. Five specimens were examined for each type of material and the mean value was reported for statistical analysis.

### 2.4. Shock Absorption Test

Combinations of each of the core materials and each of the cushioning materials were used for the shock absorption test. The shock absorption test was performed using an impact testing machine (modified IM-201, Tester Sangyo Co., Saitama, Japan) (Figure 3). The impact load was applied to the test specimen by dropping a weight of 500 g from a height of 250 mm onto a steel rod with rounded ends with a diameter of 3/16th of an inch. Two different measuring systems were used: a load cell sensor system and a film sensor system.

The impact load was measured using three dynamic compression load cells (LMB-A-2KN, Kyowa Electronic Instruments Co., Tokyo, Japan), which were placed in a triangle below a 10-mm-thick stainless steel platform. The specimen was placed at the center of this platform. During the test, the load was recorded using data analysis software (EDX-100A and DCS-100A, Kyowa Electronic Instruments Co.) at a sampling rate of 20 kHz. The total impact load was calculated as the sum of the readings of the three load cells, and the maximum load after the application of the impact load was defined as the maximum load. The results obtained without a test specimen were used as the references.

For the pressure distribution measurements, a pressure measurement film (Presheet, Fujifilm Corp., Tokyo, Japan) was placed under the test specimen. Recommended pressure ranges over 2.5 MPa (LW) sensitivity films were used. The pressed regions of the film showed red discoloration depending on the pressure experienced. The pressure distribution and maximum pressure were analyzed using an image analysis system (camera system: Data Shot FPD-100 and analysis software: Data Shot FPD-100S; Fujifilm Co.). Each specimen was tested five times and the mean value was reported for statistical analysis.

### 2.5. Air Permeability Test

The specimens in each group were evaluated using Digital Frazir Type Air Permeability Tester (DAP-360, Daiei Kagaku Seiki MFG, Co. LTD, Kyoto, Japan). For the measurements, the test specimen was clamped over the test head opening by pressing down the clamping arm, which started the vacuum pump automatically (Figure 4). The air permeability test (JIS L 1096: 2010) was performed using the fabric and knitted fabric testing methods, which allowed air (flow rate in cm^3^/cm^2^·s) to pass through the test specimen. The specimens in all 12 groups as well as the two cushioning materials were tested five times each and the mean value was reported for statistical analysis.

### 2.6. Statistical Analysis

The obtained results were analyzed using two-way analysis of variance in terms of two factors (core material and cushioning material). We also performed Tukey’s honestly significant difference test and Dunnett’s test. The analyses were performed using a statistical software (SPSS Ver. 25.0 IBM, Chicago, IL, USA) at a significance level of 5%.

The flexural strengths of the ABS and AP specimens as determined by the three-point bending test were compared using the unpaired *t*-test at the 5% significance level.

## 3. Results

### 3.1. Weight Measurements

The weights of the various specimens are listed in Table 3. The weight of the conventional material, namely, AP0 (36.2 g) was greater than those of the ABS specimens. Moreover, the weight of NEO (8.5 g) was also greater than that of the AKE cushioning material (5.7 g).

### 3.2. Three-Point Bending Test (Flexural Strength)

No fracture was observed in any of the specimens during the tests. The flexural strength of ABS0 was 77.1 ± 0.8 MPa while that of AP0 was 26.8 ± 0.7 MPa (*p* < 0.001) in Figure 5. Thus, the flexural strength of ABS0 was significantly greater (nearly three times higher) than that of AP0.

### 3.3. Maximum Load (ML) during Impact Test

The ML without a test specimen was 5574 ± 131 N. Moreover, after a specimen was inserted, the ML decreased, as shown in Figure 6. When the core material was kept the same and the cushioning material was changed, the ML values were not significantly different. However, they were slightly higher when AKE was the cushioning material as compared with those in the case of NEO (*p* = 0.232). The ML in the case of the conventional combination of AP0 + NEO was not significantly different from those for the other samples, with the exception of ABS10 + NEO, ABS20 + AKE, and ABS20S + AKE. The ML for ABS10S + AKE was significantly different from those for ABS10 + AKE, ABS20S + AKE, ABS20 + AKE, ABS20 + NEO, and ABS10 + NEO. On the other hand, there was no significant difference between the two cushioning materials (AKE and NEO). However, there were statistically significant interactions between the effects of the cushioning materials and core materials in terms of the ML values (*p* < 0.001).

### 3.4. Pressure Distribution under FG

The maximum stress and impressed stress distribution areas of the specimens were evaluated based on the results of the analysis of the pressure measurement films (Figure 7, Figure 8 and Figure 9). The histograms (Figure 10) show the pressure distributions in the tested specimens.

For the same core material, the maximum stress and impressed stress distribution area were both higher when AKE was used as the cushioning material as compared with those for NEO (*p* < 0.001).

When AKE was the cushioning material, for each combination of the four types of perforated ABS materials, neither the maximum stress nor the impressed stress distribution area was significantly different from that for AP0 + NEO.

### 3.5. Air Permeability Test

Table 4 shows the air flow rates (cm^3^/cm^2^·s) of the various specimens. The conventional cushioning material NEO exhibited almost no air permeability. On the other hand, AKE showed good air permeability, which was obviously significantly greater than that of NEO (*p* < 0.001). Finally, the diameter of the holes did not affect the air flow rate when AKE was used as the cushioning material.

### 3.6. Relationship between ML and Air Permeability

The regression model for the relationship between ML and the air permeability (evaluated in terms of air flow rate) for five datasets (ABS0 + AKE, ABS10S + AKE, ABS10 + AKE, ABS20S + AKE and ABS20 + AKE) was found to be statistically insignificant (*p* = 0.219). In contrast, the regression model consisting of four datasets (ABS0 + AKE, ABS10 + AKE, ABS20S + AKE and ABS20 + AKE) was found to be statistically significant (*p* < 0.05). ABS10S does not fall within the 99% prediction interval for the regression model based on the four datasets (Figure 11).

## 4. Discussion

For this study, we selected AKE64140 as the 3D-knitted fabric because previous reports suggest that the shock absorption ability of this material is similar to that of NEO while those of other 3D-knitted fabrics are inferior [42].

The three-load-cell system used in this study could monitor the load transmitted under the FG test material over time, thus allowing the magnitude of the change in the impact load to be recorded [43]. The impact force required for maxillofacial bone fractures is 4930–5780 N [44]. In the present study, the force generated by an object (500 g) free falling from a height of 250 mm (5574 ± 131 N) was used as the impact load. The impact load absorption ability, which is the ratio of the decrease in the impact load because of the material used to the original impact load, has often been used as a parameter for evaluating FG materials [21,43,45]. The impact load absorption capability of the medical splint material used in this study was determined to be 85–88% in a previous study, which used an impact load system similar to that employed in the present study [43]. Moreover, this range is similar to that obtained in the present study (85–89%). The impact absorption capability of ABS0 + AKE was 89%, which indicated that this combination was superior to that consisting of the conventional materials AP0 and NEO (i.e., AP0 + NEO) (87%). The impact absorption abilities of the core materials with holes were 85–87%. The ML of the ABS10S (2.52 mm rounded holes) and ABS20S (2.55 mm rounded holes) specimens with NEO as the cushioning material were lower than those of the ABS10 (3.64 mm rounded holes) and ABS20 (3.57 mm rounded holes) specimens. These results suggest that smaller holes in the specimens resulted in a greater shock absorption ability. This means that reducing the diameter of the holes, even if this results in an increase in the number of holes, is preferable to increasing the diameter of the holes and reducing their number. Some of the specimens consisting of perforated ABS and AKE exhibited good shock absorption abilities as compared with that of AP0+NEO, with ABS10S + AKE exhibiting the strongest shock absorption ability among all the samples consisting of perforated ABS.

The three-point bending test was performed to determine the flexural strength. The flexural strength is the maximum stress achieved during the test and is indicative of how much stress can be applied before fracturing occurs. The ABS materials exhibited greater flexural strengths than that of the AP material. A core material with a higher flexural strength will diffuse impacts more effectively [17,46]. This was confirmed by the results of the ML measurements in this study. Thus, in terms of mechanical properties, 3D-printed ABS materials would be preferable as the core material for FGs as compared with a conventional material such as AP. The air flow rate through a perforated specimen is proportional to the open area, which in turn is proportional to the square of the radius of the holes. In contrast, the decrease in the bending (flexural) strength of a specimen caused by drilling a circular hole in the specimen is proportional to the cube of the radius of the hole. Therefore, for a given open area, the strength of a perforated sample can be optimized by decreasing the radius of the holes and increasing their number. This can be applicable in estimating the radius of the holes and the bending strength of the core material in the results of this study [47]. According to the concept of equivalent solid material for design analyses of perforated materials, the equivalent strength of the perforated material is used in place of the strength of the solid material. By evaluating the effect of the perforations on the yield strength of the material, the equivalent yield strength of the perforated material can be obtained as a function of the yield strength of the solid or unperforated material [48]. However, further study might be necessary, as this relationship is not as simple and proportional for plate-like materials as it is for rod-like materials.

The pressure measurement films used could precisely record the impact area based on the changes in its color to red [49]. The stress magnitude was analyzed based on the color density using a camera, scanner, and image analysis software. The pressure measurement films used in this study had a limited range of sensitivity, and an LW film sensor type was employed. Generally, the specimens with a lower maximum stress exhibited a smaller impressed area. An FG material with a lower maximum stress is preferable for protecting the injured area. The maximum stress and impressed area of the 3D-printed ABS core materials were smaller than those of the conventional AP core material. This was true irrespective of the cushioning material used (AKE or NEO). These results suggest that all the 3D-printed ABS materials tested, that is, those with and without holes, have shock absorption abilities that make them suitable for use in FGs.

The air permeabilities of the materials used in FGs have a determining effect on the wearing comfort of the FGs. They determine how readily air can flow through the materials. If there is no possibility of air flow through the pores of the fabric or if the flow is hindered, it will cause discomfort to the wearer [50]. For this reason, air permeability is considered an important factor for improving the wearing comfort of FGs. The air permeability of the NEO cushioning material (0.2 cm^3^/cm^2^·s) was almost zero and obviously significantly lower than that of AKE (178.9 ± 1.6 cm^3^/cm^2^·s). Therefore, the AKE cushioning material was more suitable than NEO. The monofilaments that interlink and brace the outer shell and the inner liner faces of the structure of AKE, which is like a honeycomb, were designed to ensure that it is highly resilient and exhibits outstanding impact resistance and high elastic recovery. As a result, large amounts of air can flow readily in all directions through the 3D structure of AKE (honeycomb structured) even when it is placed under the core sheet material of an FG. Therefore, the air permeability of AKE was high even when it was placed under a sheet without perforations (ABS0 and AP0: almost 70 cm^3^/cm^2^·s) and very high when it was placed under a sheet with 10% open area (ABS10S and ABS10: almost 2/3rd the air flow rate in the case of AKE alone). These results confirmed the usefulness of AKE as a cushioning material for FGs.

Figure 11 shows the relationships between the maximum load and air permeability (air flow rate) of the core materials while using AKE as the cushioning material. Although the ABS20 + AKE and ABS20S + AKE specimens showed better air permeabilities than those of the other specimens, the maximum loads of ABS20 + AKE and ABS20S + AKE were higher than that of AP0 + NEO. Therefore, ABS20 + AKE and ABS20S + AKE would not be suitable for use in FGs. On the other hand, ABS10S + AKE would be suitable for use in FGs because of the shock absorption ability and air permeability of this combination were better than those of the sample consisting of the conventional materials, that is, AP0 + NEO.

The weight of the ABS10S sample (27.6 g) was approximately 24% lower than that of the conventional core material AP0 (36.2 g). On the other hand, the weight of the AKE cushioning material was approximately 33% lower than that of NEO. In a previous study, 45 and 13 g of AP and NEO (inner surface only), respectively, were used when fabricating an FG for a facial bone fracture [51]. In contrast, ABS10S and AKE can be used in amounts of 34 and 9 g, respectively, thus resulting in a weight reduction of 15 g (26%). Thus, the ABS10S + AKE combination, which showed an adequately high shock absorption ability, can result in significant weight reduction and improve the wearing comfort of FGs as compared with the case for conventional FGs.

Finally, because the experimental materials explored in this study are expected to reduce the discomfort associated with wearing FGs, clinical assessments of the 3D-printed FGs based on these materials are necessary to determine the wearer’s satisfaction.

Although we 3D-printed ABS polymer sheets in the laboratory for our study, large-scale fabrication of FGs based on our designs can also be carried out by other processes such as molding (thermoforming and cutting), which can potentially save time and labor compared to those involved for 3D-printing. However, further studies are required to determine the actual costs of the manufacturing processes.

## 5. Conclusions

Since sports face guards (FGs) are frequently uncomfortable to wear and degrade players’ performance, the scope of this study included obtaining a new constitutive material capable to better balance the mechanical properties of FGs with the feeling of wearing them. 3-D digital technology was considered and different combinations of 3D-printed polymer sheets and 3D-knitted fabric cushioning with honeycomb structures were investigated. In this study, perforated and nonperforated 3D-printed ABS materials were combined with AKE as a cushioning material; these were compared with the conventional core material, AP combined with NEO as the cushioning material. Examinations were performed to determine the flexural strength, shock absorption ability, weight, and air flow rate of the prepared 3-D combinations, which contained ABS sheets with five different perforation patterns.

The combination of ABS10S and AKE as the cushioning material showed the requisite shock absorption ability as well as higher air permeability and lower weight than the conventional materials. These results suggest that a combination of 3D-printed ABS materials and 3D-knitted fabrics with honeycomb structures can be used in FGs to improve the balance of wearing comfort and material mechanical properties for athletes.

## Figures and Tables

**Figure 1 polymers-13-01879-f001:**
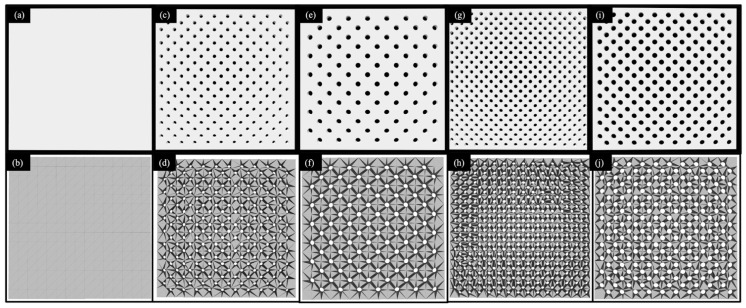
STL file view and that with frameworks of Specimens (100 × 100 mm^2^) (**a**) ABS0 (STL file), (**b**) ABS0 (framework), (**c**) ABS10S (STL file), (**d**) ABS10S (framework), (**e**) ABS10 (STL file), (**f**) ABS10 (framework), (**g**) ABS20S (STL file), (**h**) ABS20S (framework), (**i**) ABS20 (STL file) and (**j**) ABS20 (framework).

**Figure 2 polymers-13-01879-f002:**
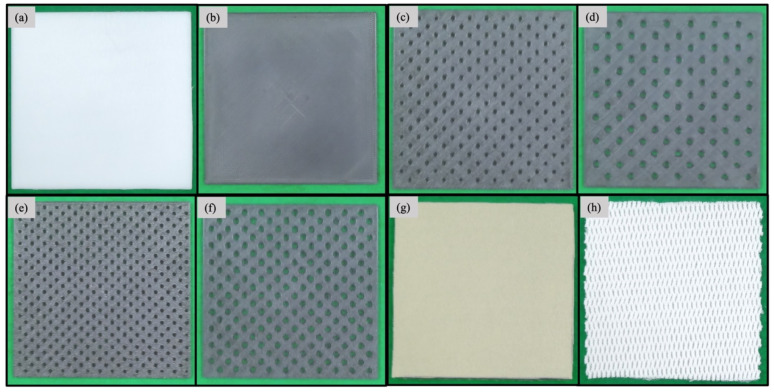
Specimens (100 × 100 mm^2^) used in this study: (**a**) AP0, (**b**) ABS0, (**c**) ABS10S, (**d**) ABS10, (**e**) ABS20S, (**f**) ABS20, (**g**) NEO, and (**h**) AKE.

**Figure 3 polymers-13-01879-f003:**
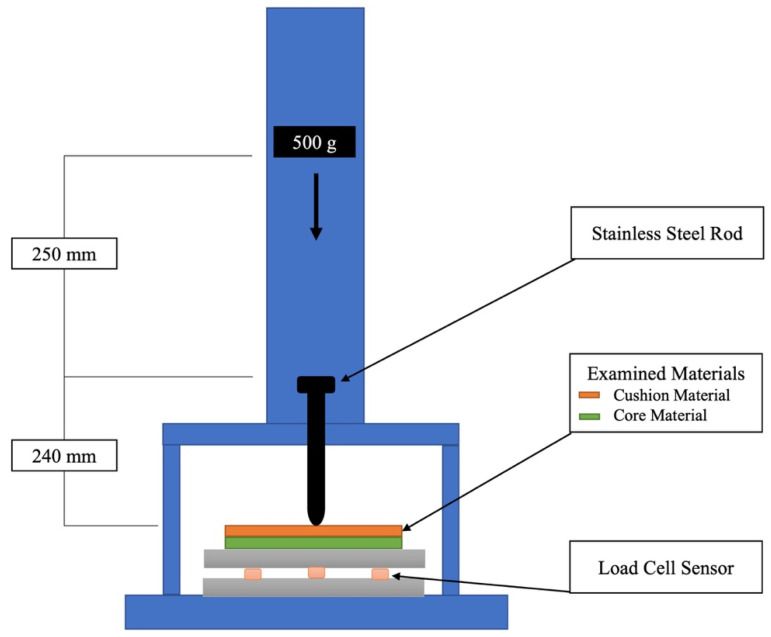
Schematic diagram of setup for shock absorption test.

**Figure 4 polymers-13-01879-f004:**
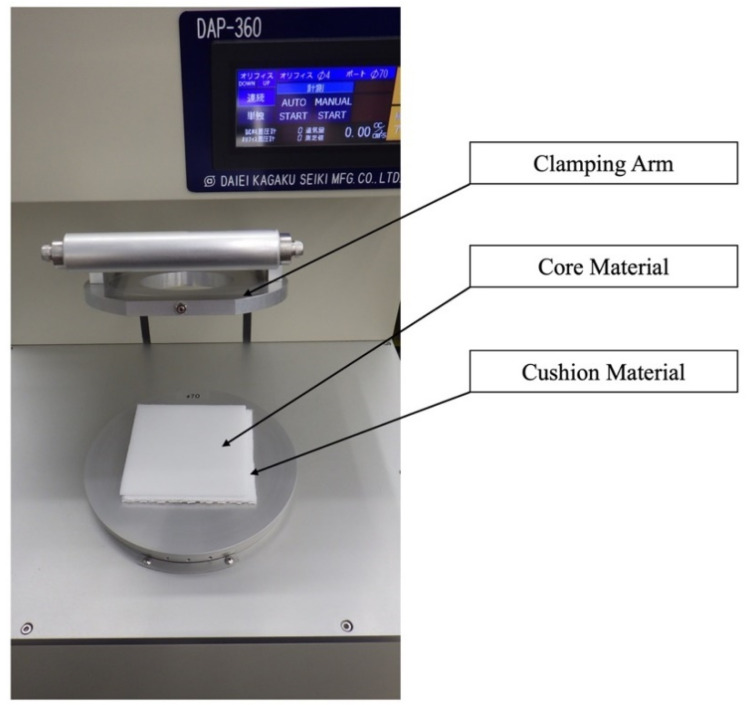
Air permeability test using a Digital Frazir Type Air Permeability Tester.

**Figure 5 polymers-13-01879-f005:**
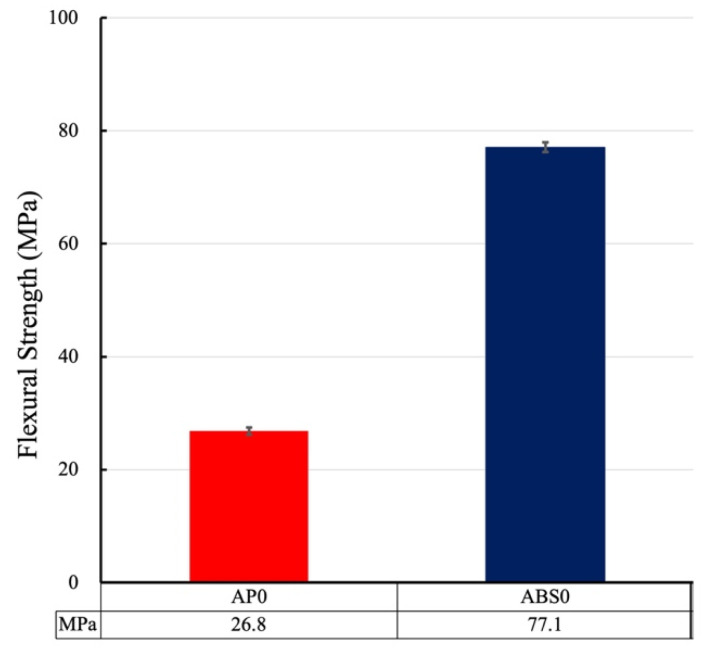
Flexural Strength of AP0 and ABS0. There was significant difference. (*p* < 0.001).

**Figure 6 polymers-13-01879-f006:**
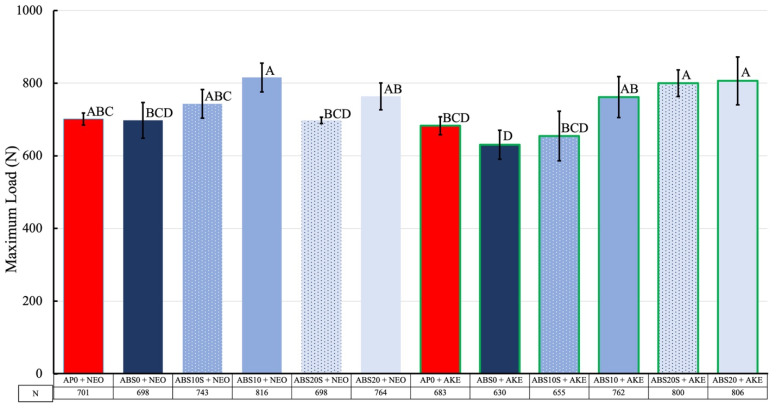
Maximum loads of various specimens obtained from the shock absorption tests. The values of the same letter were not significantly different (*p* > 0.05).

**Figure 7 polymers-13-01879-f007:**
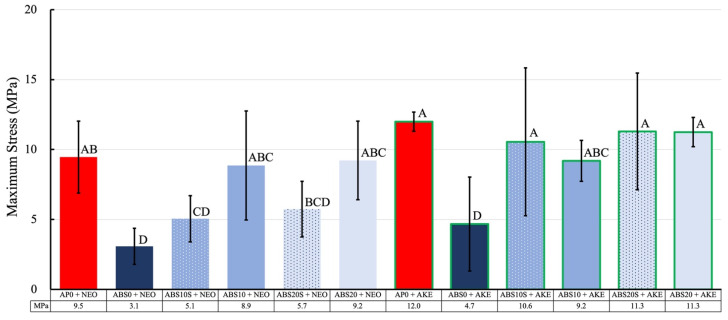
Maximum stress of various specimens obtained from the shock absorption test. The values of the same letter were not significantly different (*p* > 0.05).

**Figure 8 polymers-13-01879-f008:**
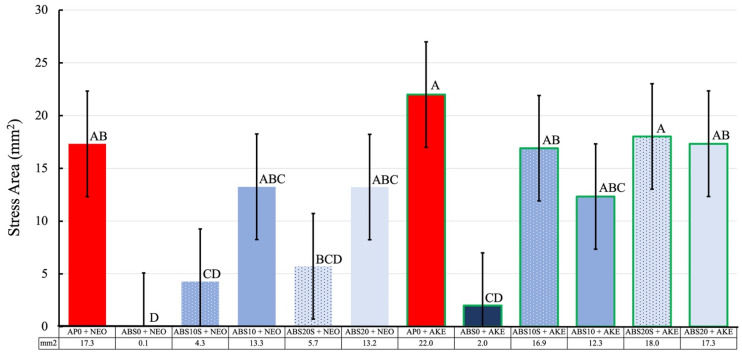
The impressed area of the maximum stress of various specimens obtained from the shock absorption test. The values of the same letter were not significantly different (*p* > 0.05).

**Figure 9 polymers-13-01879-f009:**
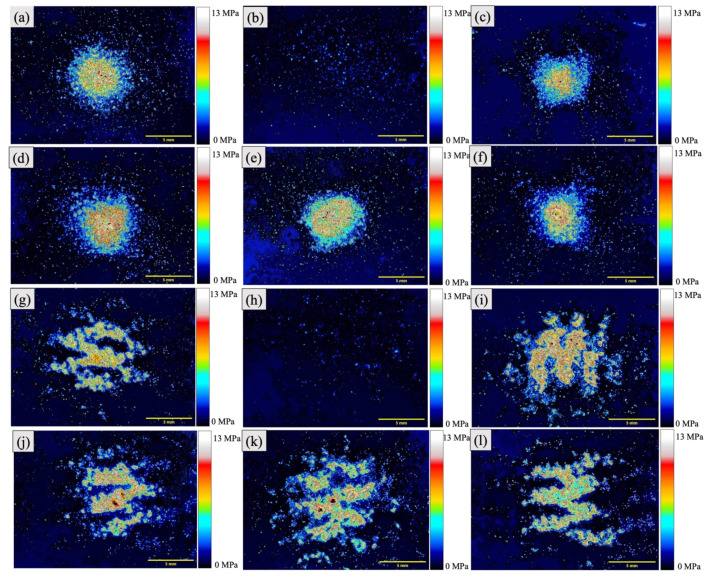
Pressure distribution obtained using pressure measurement films. (**a**) AP0 + NEO, (**b**) ABS0 + NEO, (**c**) ABS10S + NEO, (**d**) ABS10 + NEO, (**e**) ABS20S + NEO, (**f**) ABS20 + NEO, (**g**) AP0 + AKE, (**h**) ABS0 + AKE, (**i**) ABS10S + AKE, (**j**) ABS10 + AKE, (**k**) ABS20S + AKE, and (**l**) ABS20 + AKE.

**Figure 10 polymers-13-01879-f010:**
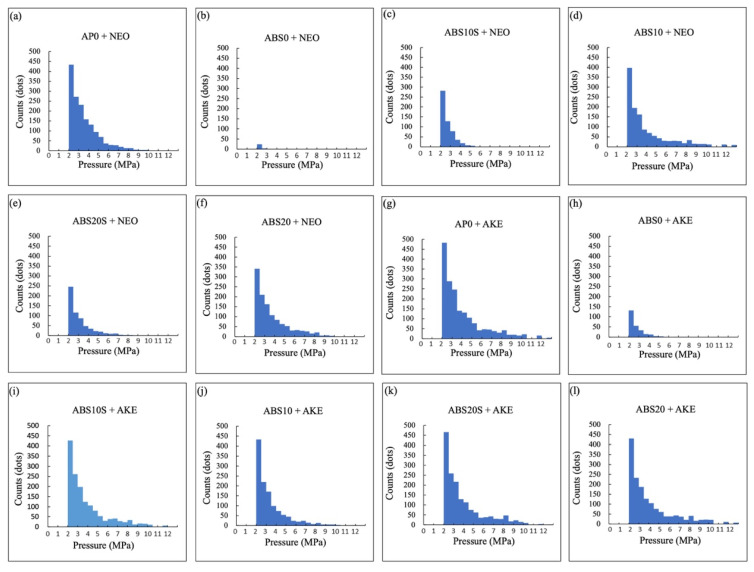
Pressure histograms obtained using pressure measurement films. (**a**) AP0 + NEO, (**b**) ABS0 + NEO, (**c**) ABS10S + NEO, (**d**) ABS10 + NEO, (**e**) ABS20S + NEO, (**f**) ABS20 + NEO, (**g**) AP0 + AKE, (**h**) ABS0 + AKE, (**i**) ABS10S + AKE, (**j**) ABS10 + AKE, (**k**) ABS20S + AKE, and (**l**) ABS20 + AKE. One dot represents an area of 0.125 × 0.125 mm^2^.

**Figure 11 polymers-13-01879-f011:**
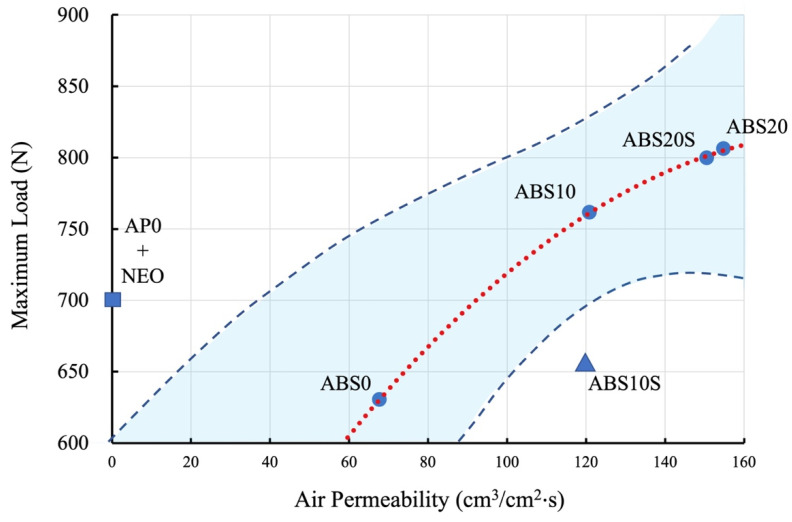
The regression model for relationships between ML and air permeability for five datasets (ABS0 + AKE, ABS10S + AKE, ABS10 + AKE, ABS20S + AKE and ABS20 + AKE) in comparison with that for conventional FG dataset (AP0 + NEO).

**Table 1 polymers-13-01879-t001:** The mesh model creation results for conversion to STL format of each specimen.

	ABS0	ABS10S	ABS10	ABS20S	ABS20
Triangle	1140	892,108	300,604	1,133,968	611,892
Apex	3420	2,676,324	901,812	3,401,904	1,835,676
File size (kB)	57	44,605	15,030	56,698	30,594

**Table 2 polymers-13-01879-t002:** Groups of examined materials in this study.

Sample Group	Core Materials					Cushioning Materials	
	Materials & Thickness	Composition & Manufacturer	Open Area %	Hole Size		Materials & Thickness	Composition & Manufacturer
AP0 + NEO	AP (3.2 mm)	Polycaprolactone (Homecraft Rolyan)	0		-	NEO (4.5 mm)	Polychloroprene with nylon lining (Homecraft Rolyan)
ABS0 + NEO	ABS (3 mm)	Acrylonitrile, butadiene, and styrene (NCI Sales)	0		-		
ABS10 + NEO			10		3.64 mm rounded		
ABS10S + NEO				S	2.52 mm rounded		
ABS20 + NEO			20		3.57 mm rounded		
ABS20S + NEO				S	2.55 mm rounded		
AP0 + AKE	AP (3.2 mm)	Polycaprolactone (Homecraft Rolyan)	0		-	AKE (4.3 mm)	3D-knitted fabric (Asahi Kasei)
ABS0 + AKE	ABS (3 mm)	Acrylonitrile, butadiene, and styrene (NCI Sales)	0		-		
ABS10 + AKE			10		3.64 mm rounded		
ABS10S + AKE				S	2.52 mm rounded		
ABS20 + AKE			20		3.57 mm rounded		
ABS20S + AKE				S	2.55 mm rounded		

**Table 3 polymers-13-01879-t003:** Weights of various specimens (100 × 100 mm^2^ sheet).

Materials	AP0	ABS0	ABS10S	ABS10	ABS20S	ABS20	NEO	AKE
weight (g)	36.2 ± 0.1	30.5 ± 0.1	27.6 ± 0.1	27.4 ± 0.1	25.5 ± 0.1	24.8 ± 0.1	8.5 ± 0.1	5.7 ± 0.1

**Table 4 polymers-13-01879-t004:** Air flow rates of various specimens obtained from air permeability test. The values of the same letter were not significantly different (*p* > 0.05).

	without Core Material	AP0	ABS0	ABS10S	ABS10	ABS20S	ABS20
NEO	0.2 ± 0.0	0.2 ± 0.0 ^A^	0.2 ± 0.0 ^A^	0.2 ± 0.0 ^A^	0.2 ± 0.0 ^A^	0.2 ± 0.0 ^A^	0.2 ± 0.0 ^A^
AKE	178.9 ± 1.6	69.0 ± 0.4 ^C^	67.7 ± 0.6 ^B^	121.8 ± 1.0 ^D^	120.9 ± 0.9 ^D^	150.6 ± 0.7 ^E^	154.8 ± 0.9 ^F^

(unit: cm^3^/cm^2^·s).

## Data Availability

The data that support the findings of this study are available from the corresponding author (T.U.) upon reasonable request.

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
