# Peer review of "Air Permeability, Shock Absorption Ability, and Flexural Strength of 3D-Printed Perforated ABS Polymer Sheets with 3D-Knitted Fabric Cushioning for Sports Face Guard Applications"

_polymers, 2021, doi:10.3390/polym13111879_

Round 1

Reviewer 1 Report

On request of Polymers, I have revised the manuscript titled “Air Permeability, Shock Absorption Ability, and Flexural Strength of 3D-Printed Perforated ABS Polymer Sheets with 3D-Knitted Fabric Cushioning for Sports Face Guard Applications” by Thet Khaing Aung et al.

Since sports face guards (FGs) frequently are uncomfortable to wear and lower players’ performance, the scope of this study was to obtain a new constitutive material capable to better balance wearing feel and mechanical properties of FGs. To this end, the authors thought about using 3-D digital technology and investigated different combinations of 3D-printed perforated acrylonitrile butadiene styrene (ABS) polymer sheets and 3D-knitted fabrics cushioning with honeycomb structures. Examinations have been performed to determine the flexural strength, the shock absorption ability, and the air flow rate of the prepared 3-D combinations which contained ABS sheets with five different perforation patterns. According to the results, the ABS sheets having 10% open area and 2.52-mm round holes, combined with knitted fabric cushioning, exhibited properties significantly improved respect to conventional materials.  

General comments

Sports face guards (FGs), usually made of a combination of polymers, including polycaprolactone sheets and polychloroprene rubber, are essential and mandatory devices worn mainly by football players or fencers, to prevent orofacial injuries. Unfortunately, the traditional polymers render them uncomfortable to wear. Stuffiness caused by poor ventilation and increased weight due to absorption of humidity from perspiration are the most issues, which also negatively affect the sporty performance. In this regard, to search for new materials for fabricating FGs could be of great utility to improve the quality of life of these sportsmen. Note that, for most of these players, their sport is their work, and are forced to wear FGs for a high number of hours during training, and to face competitions in which good performance is crucial for their career. Consequently, studies in this area are desired and welcome.  

In my opinion, the topic of the present manuscript is very interesting and, in addition to be particularly suitable for Polymers, could attract the interest of a wide audience, including also sports doctors, physiotherapists, industries, sports companies etc. Although the prepared materials need to be processed and transformed in a prototype, to assess their actual application, the early results here presented by the group of Thet Khaing Aung are very appealing and promising.

The language is good, the results well presented, and the discussion is sufficiently well organized. Collectively, the value of the manuscript is already good. However, some residual minor criticisms should be addressed. Following some suggestions to make this manuscript suitable for publication on Polymers.

In materials and methods, I suggest to add more details in Section 2.2.

Line 19-20. Badly expressed. In addition, “prevent” and “protect” have very similar significance. Hence, I suggest removing one of them. Please, consider the following revision: Sports face guards (FGs) are devices that protect athletes from maxillofacial injury, or ensure rapid return to play following orofacial damages.

Line 20. Please, remove the comma before “maxillofacial injury”. Carefully check punctuation throughout the manuscript and correct where necessary.

Lines 25-29. Too long sentence. Please, shorten it.

Line 41. Please, replace “an” with “a”.

Lines 43-45. I suggest removing this part, since the subsequent sentences repeat the same concepts.

Line 49. Please, put a dot after the Ref. [13].

Line 50. Please, replace “such that” with “so that”.

Lines 71-72. Please, replace “and” between “comfort” and “impact” with a comma.

Lines 77-78. This sentence is not clear, because it seems that the authors have found that the perforated materials show almost the same shock absorption abilities as those without holes, but Churei et al. have found it. Please, clarify the question.

Line 129. Please, specify STL at its first mention using either “Standard Triangulation Language" or "Standard Tassellation Language”.

Line 134. Please, add the city and country of the manufacturer of the 3D-Printer.

Lines 148-149. According to the Instructions for authors of MDPI journals, equations must be numbered. As an example: the following equation (1):

Flexural strength = 3Fl/2bh2                                                                         (1)

Sections 2.2-2.5. Please, specify if as results by n determinations were expressed. Were they expressed as mean of n determinations?

Lines 314 and 319. Please, put the dots after the references.

Reviewer 2 Report

Dear Authors,

The article contains interesting research results, but requires improvement of several issues:

  1. In the introduction, a fairly poor literature review was carried out on the use of 3D printing in the described field - please extend this literature review to describe the current state of knowledge based on specific 3D printing technollogues along with a description of materials. PJM technology with biocompatible materials is particularly important. I recommend publication (Design guidelines for plastic casting using 3D printing - DOI:10.1177/1558925020916037)

  2. There are no specific saving parameters in the Materials part for the stl files. Please also complete the photo of the saved STL model. Correct analysis of stl files when making such small models is extremely important - please provide a full description of saving the files with all deviations.
  3. There are no detailed print parameters and in the case of the strength analysis of the produced models it is an extremely important issue. The lack of such important parameters makes it impossible to analyze the rest of the research part.
  4. It seems that the conclusions could be extended by more than one paragraph. They are much too short to fail to analyze the remaining materials. 

Kind regards,

Reviewer

Reviewer 3 Report

Dear Authors,

The proposed manuscript is overall of an medium quality/interest with now significant issues/flaws. I enclosed some revisions which could lead to potential improvements.

Line 40: I find anomalous the use of 12 sources to motivate the statement also the citation style does not appear consistent

Line 67 a previous study is mentioned but it not cited

Figure 1 is not very significant and might be removed from the introduction section

Overall a restyling of the introduction section needs to be done. From line 75 to 99 the readers gets very confused since the topic are present in a not well structured way. All the previous bibliography should be firstly introduced and only at the point the scope and the novelty of the paper needs to be stated. Also the authors discuss the importance of computer aided design but did not stated how it is relevant to the study in the final part of the introduction section.

Equations appears to not be numbered

I suggest the authors to add a table to insert the details about the produced specimen it would be o significant help to the reader and integrate figure 2

In figures 6-7-8 the only the upper limit of the error bar is present, why is the error bar one sided ?

Round 2

Reviewer 2 Report

Dear Authors,

I accept publication.

Kind regards,

Reviewer